# Roughing Milling with Ceramic Tools in Comparison with Sintered Carbide on Nickel-Based Alloys

**Pablo Fernández-Lucio** [1,*], **Octavio Pereira Neto** [1,2], **Gaizka Gómez-Escudero** [1], **Francisco Javier Amigo Fuertes** [2], **Asier Fernández Valdivielso** [2] **and Luis Norberto López de Lacalle Marcaide** [1,2]

1   Department of Mechanical Engineering, University of the Basque Country (UPV/EHU),
    Plaza Torres Quevedo s/n, 48013 Bilbao, Spain; octaviomanuel.pereira@ehu.eus (O.P.N.);
    gaizka.gomez@ehu.eus (G.G.-E.); norberto.lzlacalle@ehu.eus (L.N.L.d.L.M.)
2   CFAA, University of the Basque Country (UPV/EHU), Parque Tecnológico de Zamudio 202,
    48170 Bilbao, Spain; famigo001@ikasle.ehu.eus (F.J.A.F.); asier.fernandezv@ehu.eus (A.F.V.)
*   Correspondence: pablo.fernandezd@ehu.eus; Tel.: +34-94-601-3932

**Abstract:** Productivity in the manufacture of aircrafts components, especially engine components, must increase along with more sustainable conditions. Regarding machining, a solution is proposed to increase the cutting speed, but engines are made with very difficult-to-cut alloys. In this work, a comparison between two cutting tool materials, namely (a) cemented carbide and (b) SiAlON ceramics, for milling rough operations in Inconel® 718 in aged condition was carried out. Furthermore, both the influence of coatings in cemented carbide milling tools and the cutting speed in the ceramic tools were analysed. All tools were tested until the end of their useful life. The cost performance ratio was used to compare the productivity of the tested tools. Despite the results showing higher durability of the coated carbide tool, the ceramic tools presented a better behavior in terms of productivity at higher speed. Therefore, ceramic tools should be used for higher productivity demands, while coated carbide tools for low speed-high volume material removal.

**Keywords:** SiAlON; cemented carbide; milling tool; Inconel® 718; coatings; MRR





## 1. Introduction

Just before the COVID pandemic (CV19), the aeronautical industry was constantly growing. Looking back a decade, it is observed that the global aircraft fleet has increased exponentially, reaching a total increase of more than 20%. In fact, the forecasts of the two big aircrafts manufacturers concluded that, in the next twenty years, the growth of the number of airplanes would be similar to the current fleet. In the mid-term and after CV19 will be solved, the industrial concern will be to reduce emissions and all kind of harmful manufacturing sources, such as coolants, or reducing energy consumption [1].

Added to this, half of the current global fleet is going to be substituted. With this, the new deliveries expected for the end of the decade of 2040 will overcome the aircraft fleet of 2018 [2,3]. However, the use of super alloys, commonly used in aircrafts engines, implies a very difficult task to reduce machining times [4,5]. Therefore, machining processes must be more productive to achieve those deliveries. In the last century, three main cutting tool developments were carried out. First of all, the emergence of the cemented carbide tools, followed by the introduction of CVD (physical vapor deposition) and PVD (physical vapor deposition) coatings in the industry, and finally, the appearance of ceramic grades.

Cemented carbides are composite materials, made of hard WC (Tungsten carbide) grains in a ductile matrix of cobalt [6]. In fact, cemented carbide tools were initially developed in the 1930s. With the introduction of these kinds of tools in the industry, cutting speed and tool life increased drastically in comparison with high-speed steels, developed around the early years of the 20th century.

Nevertheless, due to the rapid notch wear caused by the chemical affinity between carbon and the austenitic phase of steels, new types of carbides, such as TaC, NbC, and TiC, were added. With these additional elements, it constitutes a three-phase cemented carbide more suitable for steel machining [7,8]. It was not until the end of the 1970s that the cemented carbide tools were generalised. This happened due to machine evolution in that decade, when computer numerical control (CNC) became a common machine standard.

Nonetheless, with the development of coatings for cutting tools at the end of the 1980s, the cemented carbides uncoated were sent to a second plane [9]. In fact, with a small coating layer up to 12 μm, tool life improved drastically even cutting speed is increased [10–13]. Moreover, coatings prevent tools from oxidation and corrosion, so tools reached higher cutting speeds [14].

With the appearance of modern technologies, coatings advanced from a single layer to multiple layers. In this way, different coatings combinations, such as TiN, TiCN, TiC, or $Al_2O_3$, could be made to improve cemented carbide tools properties [15,16]. In fact, with the correct combination of coatings, tool life increases more than each one of the coats separated [17].

However, cemented carbides cannot deal with heat-resistant alloys at high cutting speeds. Thus, the development of ceramic tools supposed an improvement in super alloys machining performance, enhancing productivity [18]. This is due to the fact that ceramic tools allow to achieve high cutting speeds thanks to their hardness and good abrasion resistance [19,20].

In turning operations, where machining processes are very stable and continuous without much impact on the cutting edge, both material removal ratio (MRR) and ceramic inserts life are higher than cemented carbide tools [21]. Indeed, ceramic tools are the perfect solution to machine difficult-to-cut materials, such as austempered ductile iron (ADI) castings [22]. Nevertheless, in high-speed milling, there is not much research due to the fact that in milling operations the cutting action is interrupted, and therefore harmful for ceramic tools.

In this line, only a few researches can be read in the literature. Among them, it should be noted the work carried out by Wang et al. in 2016 [23] and Çelik et al. in 2017 [24], respectively. In the first one, it was stated that during H13 steel milling with finishing cutting conditions, ceramic tool lives were longer and cutting forces lower in comparison with carbide tools [23]. In the other one, ceramic tools were used to finish operations in Inconel® 718. They concluded that ceramic tools increased productivity and their tool life was higher in comparison with cemented carbide tools. Nonetheless, the surface integrity was affected what implied an unsuitable process [24]. Then, despite achieving higher tool lives, ceramic tools cannot be used for finishing operations in industrial environments.

Recent research has aimed at new ways of lubrication and cooling, such as high-pressure and cryogenics, the latter using $CO_2$ with or without minimum quantity of lubricant. Suárez et al. stated that high-pressure coolant reduces cutting forces and tool wear [25]. Therefore, productivity in the machining of heat-resistant alloys was increased, but with a higher environmental impact. The use of cryogenic with minimum quantity of lubricant in the machining of nickel-based alloys can be presented as a feasible replacement for wet machining in terms of the lower environmental impact [26,27].

Assisted processes were also tested, by using ultrasonic or thermal sources, such as plasma. In their study, López de Lacalle et al. concluded that the use of this technique during the machining of heat-resistant alloys with an alumina reinforced with CSi whiskers tool increases tool life and reduces cutting forces [28]. Plasma assisted machining was applied using ceramic tools, so in future it can be a consequent line for this work.

Moreover, new techniques were used to deal with nickel-based alloys and enhance productivity. González et al. analysed the use of super abrasive machining as a substitute of flank milling roughing operations. They concluded that super abrasive machining in heat resistant alloys was a suitable process to improve roughness, hardness, and residual stresses in manufactured components [29].

Taking these issues into account, in this work milling Inconel® 718 using SiAlON ceramic milling tool in roughing operations under high-speed milling conditions is carried out. The main target of this research is to analyse tool life in this kind operations in comparison with cemented carbide tools uncoated and coated, respectively. The results show that, in roughing operations, the MRR with ceramic tools is higher, and therefore an increase of productivity is achieved as well as a lower environmental impact.

## 2. Materials and Methods

In order to analyse the difference between roughing operations with cemented carbides and ceramics tools in a heat-resistant super alloy, peripheral milling with different kinds of tool materials were carried out. According to ISO-8688-1/1989 [30], the final test criteria were established when either average tool wear reached 0.3 mm or a fatal failure of the tool was produced. Between each test, a different cutting tool was used to clean the workpiece from the previous milling test. Each test was repeated three times, using a new tool for each repeated test.

The material used during the tests was Inconel® 718. This heat resistant super alloy is characterized by its high resistance to fatigue, creep, and corrosion under extreme working conditions at high temperature [31]. Therefore, it is used widely in aeronautical turbomachinery critical components [32]. Table 1 presents the chemical composition and mechanical properties of Inconel® 718 provided by the supplier, here in heat treatment known as aged.

**Table 1.** Inconel® 718 chemical composition and physical properties of tested alloy.

| Chemical Composition (%) | | | | | | | | | | | | |
|---|---|---|---|---|---|---|---|---|---|---|---|---|
| Ni | Cr | Co | Fe | Nb | Mo | Ti | Al | B | C | Mn | Si | Others |
| 52.5 | 19 | 1 | 17 | 5 | 3 | 1 | 0.6 | 0.01 | 0.08 | 0.35 | 0.35 | 1.79 |

| Mechanical Properties | | | | | | |
|---|---|---|---|---|---|---|
| Hardness | Young's Modulus | Tensile Strength | Density | Specific Heat | Melting Temp. | Thermal Conduct. |
| 42 HRC | 206 GPa | 1.73 GPa | 8470 kg/m³ | 461 J/(kg·K) | 1550 K | 15 W/(m·K) |

However, due to the excellent mechanical properties for its working environment, Inconel® 718 is considered a difficult-to-cut alloy. Inconel® 718 combines high hardness with high ductility which produces low material removal ratios, built-up edges, extreme tool wear and high cutting forces [33,34]. Therefore, it is a suitable material to enhance the differences between ceramics and cemented carbides behaviour under extreme machining conditions.

Peripheral millings were carried out in a milling centre IBARMIA ZV-25/U600 (Ibarmia, Azkoitia, Spain) with Inconel® 718 aged workpieces of $250 \times 100 \times 5$ mm³. To analyse the wear evolution, a PCE-200 microscope (PCE Holding GmbH, Hochsauerland, Germany) was used. Figure 1 shows the experimental set-up for the tests. The machine spindle was 18 kW.

In these tests, three different kinds of tool with a diameter of 12 mm, four teeth, and the same geometry were used. The helix angle of the tools was 40°. The first one was a cemented carbide uncoated tool. Another cemented carbide milling tool was also tested with AlTiN coat. This coating is suitable for machining Inconel® 718 displaying high oxidation resistance and high hot hardness among others [35,36]. The substrate of both tools was EMT612, a substrate that combines a high transverse rupture strength with high hardness. Finally, uncoated ceramic tools were tested using two different cutting speeds. These tools were composed of SiAlON that provides an excellent mechanical strength in addition to the toughness of silicon nitride [37].

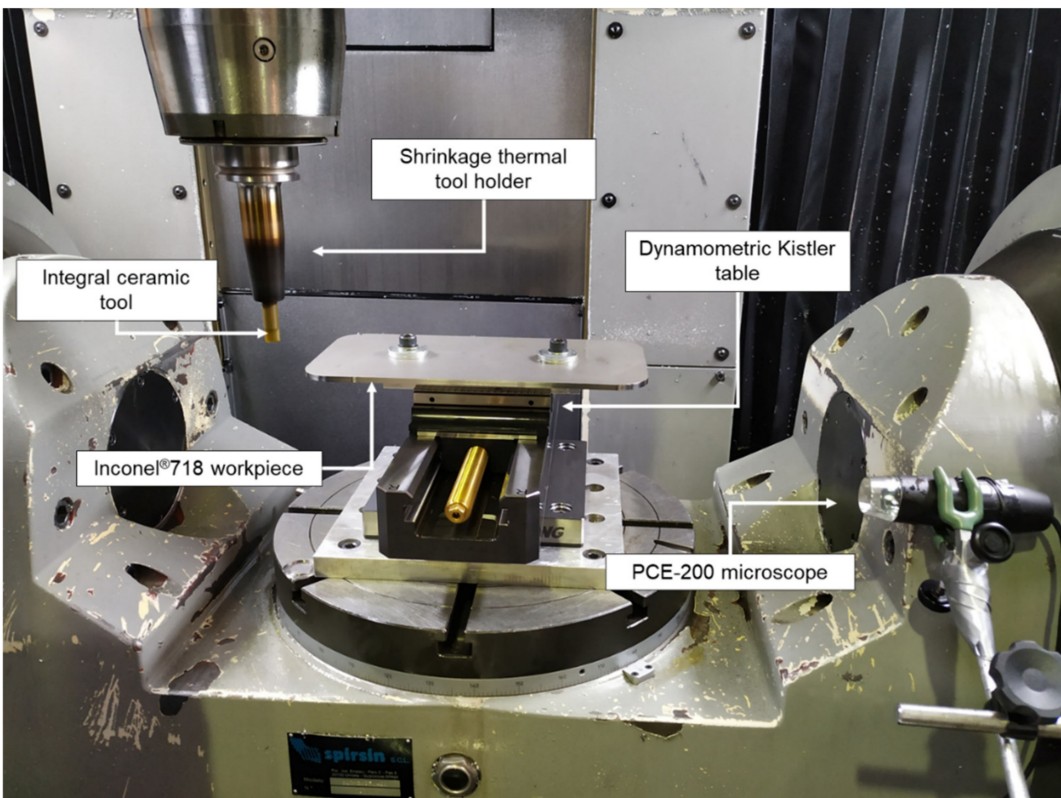

**Figure 1.** Experimental setup in the milling centre.

It should be noted that cemented carbide tool tests were carried out with oil emulsion lubrication at 10%, while the ceramic tools were tested in dry conditions. The cutting parameters were established taking into account manufacturer recommendations. Finally, in Table 2, the cutting parameters used with each tool during the tests are shown, where the price is defined in ECUs (Equivalent Cost Unit). In this case, the reference cost is established according to euro value at May 2021: 1ECU equals to 1€.

**Table 2.** Cutting parameters selected for the milling process. Cutting speed ($v_c$), feed per tooth ($f_z$), axial depth of cut ($a_p$), radial depth of cut ($a_e$), material removal rate (Q) and price.

| Tool | Ceramic | | EMT612 | |
|---|---|---|---|---|
| | | | AlTiN | Straight Carbide |
| | 1A | 1B | 2 | 3 |
| $v_c$ [m/min] | 680 | 452 | 20 | 20 |
| $f_z$ [mm/z] | 0.03 | 0.03 | 0.03 | 0.03 |
| $a_p$ [mm] | 5.6 | 5.6 | 5.6 | 5.6 |
| $a_e$ [mm] | 1.0 | 1.0 | 1.5 | 1.5 |
| Q [mm$^3$/min] | 12.096 | 8.064 | 0.5544 | 0.5544 |
| Price [ECU] | 135 | 135 | 82 | 71 |

## 3. Results & Discussion

Difficult-to-cut heat-resistant superalloys preserve good mechanical properties until 700 °C, the temperature at which these properties decrease exponentially. Therefore, the objective is to achieve this high temperature. However, at this extreme condition, the

cemented carbide tools are out of their optimal cutting parameters. Zheng et al., 2016 [38] has experimentally evaluated the high-speed cutting performance of the SiAlON tools for Inconel® 718 milling. The cutting temperature were in the range of 850–1000 °C which otherwise could be highly detrimental to cemented carbide tools.

The results obtained of tool wear during the tests carried out are shown in Figure 2. In the figure it is represented the average of tool wear for each tool. In the case of both cemented carbide tools—without and with coating—there were two kinds of tool wear. In the beginning stages, flank wear was predominant. However, extended chipping appeared at the end of the test. In fact, in the coated tool the accused chipping was the cause of the premature end of its useful life due to stability problems in the machining. Besides, tool wear evolution is similar in both cases until chipping appears in the uncoated tool, this tool achieving 2000 mm of tool life and removing 16,800 mm$^3$ of material. In the case of the coated tool, its coating delayed the chipping effect until 3600 mm and removing 30,240 mm$^3$, supposing an increase of ≈80%.

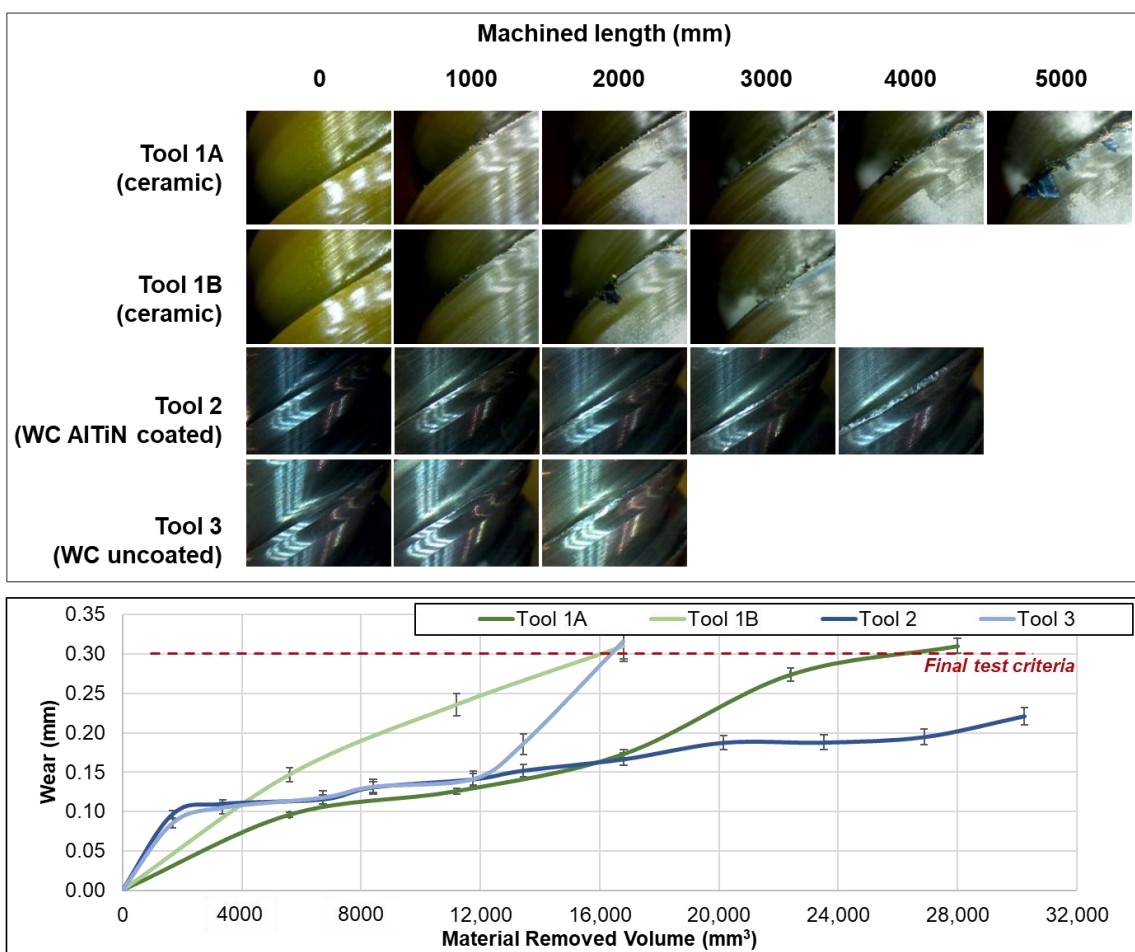

**Figure 2.** Tool wear evolution. These are the average value of three-test repetition.

In the case of both ceramic tools, they presented three types of tool wear. At the beginning of the test, both tools presented flank wear. However, at the middle of their tool lives, a built-up edge appeared, and all the teeth broke in the same part. This breakage is due to notch wear as is shown in Figure 3a. Taking into account both cutting speeds used, there is an appreciable difference just in the initial stages. Particularly, when ceramic tool was machining with 680 m/min the material volume removed was 28,000 mm$^3$ and 16,800 mm$^3$ when 452 m/min is used, achieving a total cutting length of 5000 and 3000 mm, respectively. This implied a difference between them of 66%, presenting similar tool wear. On the other hand, the machined part presented too much burr in both sides of

the perimeter as is shown in Figure 3b. This implies that they are suitable to be used only for roughing operations, necessitating the use of cemented carbide tools to carry out finishing operations.

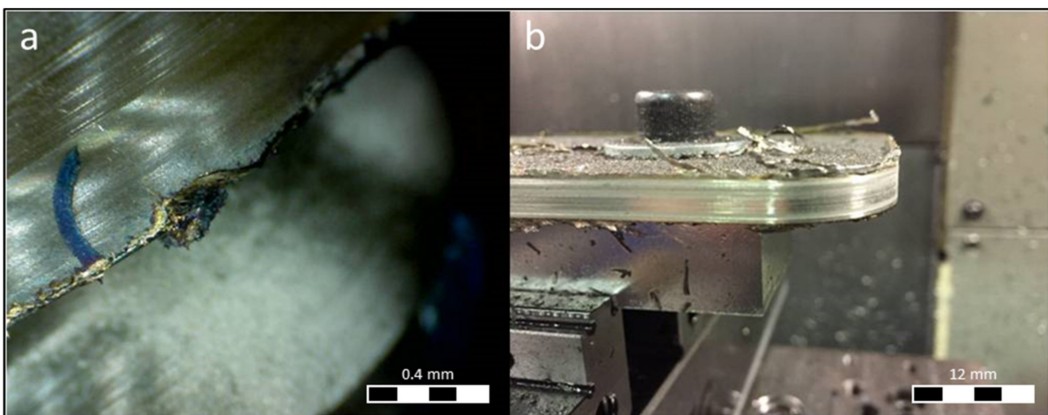

**Figure 3.** Notch wear in the ceramic tool (**a**) and surface finish of the workpiece after 5000 mm (**b**).

Regarding the material removal ratio (MRR), both tool materials present appreciable differences between them. In particular, with the WC tools the material removed was higher achieving 30,240 mm$^3$ in the case of the coated tool. However, the MRR was better with ceramic tools where 12.096 cm$^3$/min when 680 m/min was used in comparison with cemented carbide tools, which achieved 0.554 cm$^3$/min. This supposes an increase of 2082%.

To analyse the feasibility of the four tools in terms of costs, the cost performance ratio (CPR) has been used. This cost-model was introduced by Ståhl et al., 2007 with the aim of analyse the influence of the different production technologies that appear during the production development of a component [39]. Johansson et al. presented a simplified version (Equation (1)) of the initial model in which the part cost (*k*) was divided in five terms: tool costs, material costs, operator's costs, running machine costs, and idle machine cost [40].

$$k = k_{tool} + k_{material} + k_{machineON} + k_{machineOFF} + k_{operators} \tag{1}$$

In this case, only machining and tool costs have been taken into account. In Equation (2) [40] the parameters used to calculate it can be seen.

$$k = \frac{k_A * t_e}{z * T} + \frac{k_{CP} * t_e}{60 * (1 - q_{rem}) * (1 - q_{tct}) * (1 - q_Q)} \tag{2}$$

where $k_A$ is the cost of the tool, $z$ the number of cutting edges (in this case only one), $T$ tool life, $k_{CP}$ the machine costs while is running, $q_{rem}$ is rate of remaining time, $q_{tct}$ the rate of tool changes, $q_Q$ the rate of quality costs and $t_e$ the machining time. The values of $k_A$, $q_{rem}$, $q_{tct}$, and $q_Q$ values are taken from manufacturers catalogue and literature [40] respectively. Tool life ($T$) and machining time ($t_e$) are obtained from the machining tests. Machining cost per hour ($k_{CP}$) is calculated by summing the electricity consumption, coolant consumption, and machine utilization cost.

Table 3 shows the values for each parameter. To differentiate the dry cutting and the oil emulsion refrigeration, the costs of using the coolant is included in the $k_{CP}$. Moreover, in order to make a comparison between the tools, it has been supposed to machine 1000 mm$^3$.

The results of the comparison between the cutting tools using the CPR can be seen in Figure 4. The ceramic tool at higher cutting speed presented the best performance amounting to a cost of 4.92ECU. In the case of the coated cemented carbide tool, this cost ascended to 4.98ECU. On the one hand, having a look to the cost breakdown, in the case of the ceramic tool, the contribution of the tool costs is very close to the final cost, whereas

the machining costs are very low. On the other hand, it is true that the coated cemented carbide tool is cheaper. However, the machining costs are much bigger than in the ceramic case due to the use of coolant.

**Table 3.** Parameters to calculate the CPR.

| Tool | Ceramic | | EMT612 | |
|---|---|---|---|---|
| | 1A | 1B | 2 | 3 |
| $k_A$ [ECU] | 135 | 135 | 82 | 71 |
| $T$ [min] | 2.31 | 2.08 | 54.55 | 30.30 |
| $k_{CP}$ [ECU/h] | 67.06 | 66.97 | 71.15 | 71.15 |
| $q_{rem}$ [-] | 0.02 | 0.02 | 0.02 | 0.02 |
| $q_{tct}$ [-] | 0.02 | 0.02 | 0.02 | 0.02 |
| $q_Q$ [-] | 0.02 | 0.02 | 0.02 | 0.02 |
| $t_e$ [min/cm$^3$] | 0.08 | 0.12 | 1.80 | 1.80 |

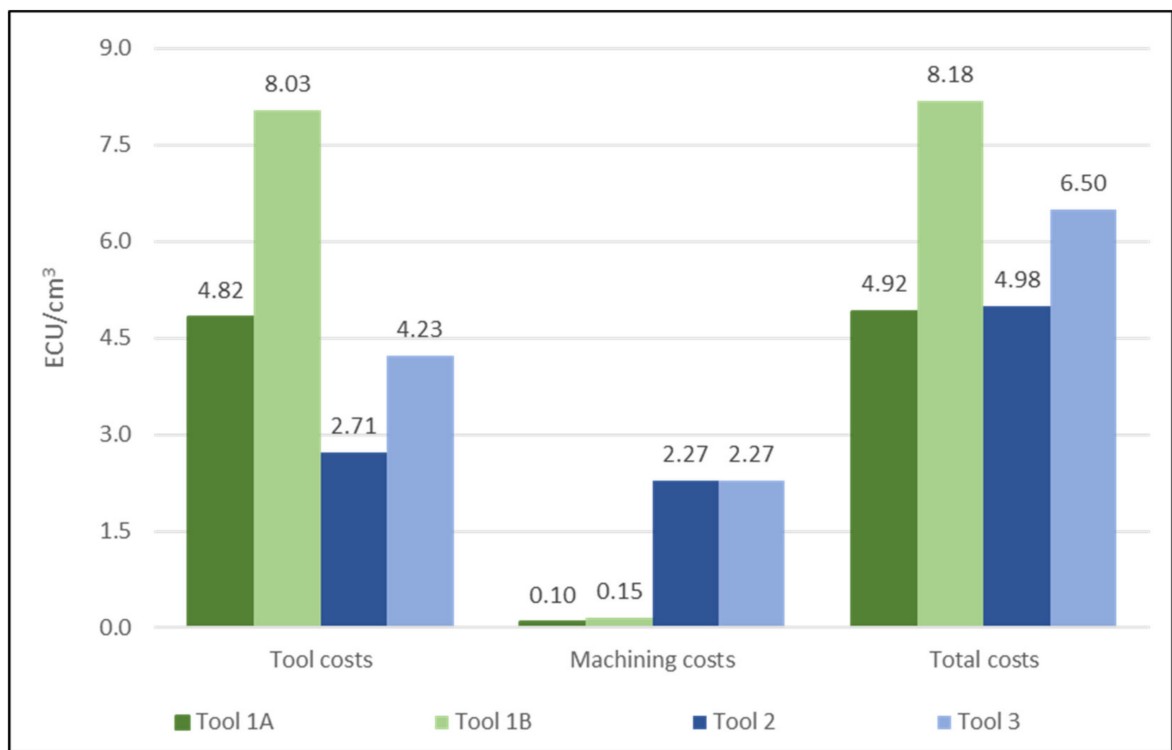

**Figure 4.** Running costs of each tool for 1 cm$^3$ machining.

Therefore, despite being more expensive, in the case of needing an increase in the productivity ceramic tools presents a suitable performance in roughing operations in comparison with WC tools. This supposes an advance in this type of heat-resistant superalloys in which an increase of productivity is mandatory, and with ceramic tools, a balance between technical and productivity issues.

## 4. Conclusions

In this work, a comparison between SiAlON ceramic and cemented carbide (uncoated and AlTiN coating) milling tools was carried out. All the tools had the same geometry. Tests were performed with cutting conditions recommended by the manufacturer and they

finished according to the final test criteria. During the tests, all wear mechanisms in both kind of tool were analysed, and the following conclusions were drawn:

1. In the case of cemented carbide tools, the coating supposes higher tool life. In fact, in these tests, the coated tool improves tool life in an 80% in comparison with uncoated tool. On the other hand, it should be noted that, when the coating disappears from the tool surface, chipping phenomenon appears and the wear increase rapidly, finishing the test.

2. In the case of ceramic tools, it was concluded that their behaviour is better when the cutting speed is higher. At 452 m/min, notching wear appeared after 3000 millimetres machined. However, increasing cutting speed to 680 m/min retarded the appearance of notching until 5000 millimetres. This implies an increase of the productivity of ≈67%.

3. The higher MRR is presented by ceramic tools. In particular, the increase is 2082% in comparison with carbide cemented tools. However, after the machining in the case of ceramic tools, the workpiece presents built-up-edge. Then, this type of tool is only suitable for roughing operations.

4. The ceramic tools with a cutting speed of 680 m/min presented the best performance in terms of cost. In particular, the cost of machining 1000 mm$^3$ with the ceramic tool was 4.92ECU while the coated cemented carbide tool was 4.98ECU.

To summarise, both kinds of tool can be useful in the industrial environment depending on the characteristics of the operation. Cemented carbide cutting tools could be used for low cutting, speed-high volume material removal whereas ceramic tools shall be recommended for higher productivity demands in machining of Inconel$^®$ 718.

**Author Contributions:** Data curation, P.F.-L., G.G.-E. and F.J.A.F.; Funding acquisition, L.N.L.d.L.M.; Investigation, P.F.-L.; Methodology, O.P.N.; Project administration, L.N.L.d.L.M.; Resources, A.F.V. and L.N.L.d.L.M.; Software, F.J.A.F.; Supervision, O.P.N. and A.F.V.; Validation, G.G.-E.; Writing— original draft, P.F.-L.; Writing—review & editing, O.P.N. and L.N.L.d.L.M. All authors have read and agreed to the published version of the manuscript.

**Funding:** This research received no external funding.

**Institutional Review Board Statement:** Not applicable.

**Informed Consent Statement:** Not applicable.

**Data Availability Statement:** Not applicable.

**Acknowledgments:** All the authors would like to thanks all the funds from Excellence groups of the Basque university system n IT1337-19, and to project Spanish Ministry of Science DPI2016-74845-R and RETOS NewMine RTC-2017-6039. Besides, the authors wish to acknowledge the financial support received from HAZITEK program, from the Department of Economic Development and Infrastructures of the Basque Government and from FEDER founds, related to the projects with acronym HARDCRAFT and TURALOY. Besides, thanks are also addressed to the Vice chancellor of innovation, social compromise and cultural action from UPV/EHU (Bizialab program from Basque Government) the UPV/EHU itself for the financial aid for the pre-doctoral grant PIF 19/96.

**Conflicts of Interest:** The authors declare no conflict of interest.

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
