# Peer review of "Roughing Milling with Ceramic Tools in Comparison with Sintered Carbide on Nickel-Based Alloys"

_coatings, doi:10.3390/coatings11060734_

Round 1

Reviewer 1 Report

Comment 1
Abstract
The authors must improve the presentation of the Abstract. The results of the paper must added.

Comment 2
Line 47
Computer numerical control
replace
Computer Numerical Control (CNC)

Comment 3
Line 66
such as ADI castings [21].
replace
such as Austempered Ductile - Iron (ADI) castings [21].

Comment 4
Line 70
Wang et al. 2016 and Çelik et al 2017, respectively.
There are not Wang et al. 2016 and Çelik et al 2017!!!

Comment 5
Table 1
The authors must give more details for Table 1 (supplier or experiment)

Comment 6
Delete the space in the page end (page 3, 5 and 6)

Comment 7
Line 128
three different tools of 12 mm with four teeth
The authors must explain if the number of three different tools are for each tool 

Comment 8
Table 2
How did the authors compare the performance for the 4 different tools 
if they did not have the same cutting conditions
(ae = 1.0 1.0 1.5 1.5)!!!

Comment 9
Line 142
It's not so good to start the sub-chapter at the bottom of the page without using text

Comment 10
Lines 146 - 147
(see Error! Reference 
source not found.) 
What is the "Error! Reference source not found."

Comment 11
Lines 147 - 148
Error! Reference source not found.) 
What is the "Error! Reference source not found."

Comment 12
Figure 2
The authors must give more details for FEM simulation.

Comment 13
Lines 149 - 150
Error! Reference source not found.) 
What is the "Error! Reference source not found."

Comment 14
Lines 155 - 156
Error! Reference source not found.) 
What is the "Error! Reference source not found."

Comment 15
Line 170
in Figure 1 (a). 
There is not Figure 1 (a)!!!!

Comment 16
Line 176
is shown in Figure 1 (b).
There is not Figure 1 (b)!!!!

Comment 17
Line 181
material removed rate (MRR),
replace
Material Removed Rate (MRR),

Comment 18
Lines 180 + 127
Two figures 1!!!!
The authors did not read the final manuscript for the final approval (too many authors).
Too many format errors. 

Comment 19
Lines 204 - 205 
Error! Reference source not found.) 
What is the "Error! Reference source not found."

Comment 20
Lines 204 - 205 
The Table 3 must be accompanied on the same page as the Table's title.

Comment 21
Line 211 
Error! Reference source not found. 
What is the "Error! Reference source not found."

Comment 22
Lines 220 - 224
Text: Full alignment 

Comment 23
Increase the number of the reference papers including (primarily) from Coatings.
The authors use 0 paper from Coatings journal / 1 paper from MDPI Journals / 32 papers from journals (References)
Τhe number for papers from MDPI journals
is considered insufficient (in reviewer's opinion).

Reviewer 2 Report

 Accept in present form

Author Response

Thank you for your time evaluating our work.

Reviewer 3 Report

Introduction:
good introduction, but you need to add more references, for example, related to the aviation industry new trends

Aviation industry trends:

Vicenzi, B.; Boz, K.; Aboussouan, L. Powder metallurgy in aerospace – fundamentals of pm processes and examples of applications. Acta Metallurgica Slovaca 2020, 26, 144-160, doi: 10.36547/ams.26.4.656

Kondo, H.; Hegedus, M. Current trends and challenges in the global aviation industry. Acta Metallurgica Slovaca 2020, 26, 141-143, doi: 10.36547/ams.26.4.763

Also, for the introduction add other refs.:

Inconel:

Brehl, D.E.; Dow, T.A. Review of vibration-assisted machining. Precision Engineering 2008, 32, 153-172, DOI: 10.1016/j.precisioneng.2007.08.003

Ridolfi, M.R.; Folgarait, P.; Di Schino, A. Modelling of laser powder bed fusion process for different type materials. Acta Metall. Slovaca 2020, 26, 7-10, doi: 10.36547/ams.26.1.525

Dudzinski, D.; Devillez, A.; Moufki, A.; Larrouquère, D.; Zerrouki, V.; Vigneau, J. Review of developments towards dry and high speed machining of Inconel 718 alloy. International Journal of Machine Tools and Manufacture 2004, 44, 439-456, DOI: 10.1016/S0890-6955(03)00159-7

Choudhury, I.A.; El-Baradie, M.A. Machinability of nickel-base super alloys: A general review. Journal of Materials Processing Technology 1998, 300, 278-284, DOI: 10.1016/s0924-0136(97)00429-9

See again table 1: density - m3

Table 2: Due you have a graph of at least one total measurement, there you are taking the average values of the measurement values? Error bars?

Deform 3D - you need to write more information related to the process parameters.

Fig. 3 mm3; comma or point?

Table 3: validity? coefficient of calculation?

Conclusions:

To summarizing, both kinds of tools can be useful in the industrial environment depending on the characteristics of the operation: Cemented carbide for a major material removed the volume of Inconel® 718 when the cutting time is not so important; ceramic tools when productivity is more important like in aeronautic turbines industry.

This statement needs to be rewritten.

Reviewer 4 Report

Some observation

in line 79 you write CO2, If you mean carbon dioxide isn't correct CO2

Line 142 "3. Results & Discussion", in my PDF is at the end of page. Try to move in the next page. 

Line 260 - References - try to us template indications for second line. For example at line 263 "(accessed on May 29, 2019)." must be under "Global Market...." 

Round 2

Reviewer 1 Report

Comment 1
Line 34
[2,3].However,
replace (insert a space)
[2,3]. However,

Comment 2
Line 36
deliveries.In the last 
replace (insert a space)
deliveries. In the last 

Comment 3
Line 68
carbide tools[21]. Indeed,
replace (insert a space)
carbide tools [21]. Indeed,

Comment 4
Line 74
Wang et al. 2016 and Çelik et al 2017, 
replace
Wang et al. [23] and Çelik et al. [24],

Comment 5
Lines 148 + 220 
Table 2 vs Table 3
Table 2: Tool  1A  - 1B  - 2  - 3
Price [ECU]    135 - 135 - 82 - 71
Table 3: Tool  1A  - 1B  - 2  - 3
Price [ECU]    130 - 130 - 82 - 71

The results from tool 1A are 135 or 130 or something else?
The results from tool 1B are 135 or 130 or something else?

Comment 6
The authors must explain what the meaning of the FEM results?
How did the authors use the FEM results according to the paper's scope?
If the FEM results must presented, insert a Table with FEM parameters.

Comment 7
Lines 214 - 215
Machining cost per hour (kCP) is calculated 
by summing the electricity consumption, coolant consumption and machine utilization cost.
The authors must be give more details how calculated the values:
67.06 - 66.97 - 71.15 - 71.15 

Comment 8
Figure 5
8.04 + 0.15 = 8.19 not 8.18
The values 4.92 - 4.98 are very close and if the value kA is not 130 or 135 or something else for 1A
the paper's results are different if the values are 130 or 135.
It is not a typical error. 

Reviewer 3 Report

Accept in present form.

Author Response

(The authors gave the same response as above.)
